# Effect of Copolymer on the Wrinkle Structure Formation and Gloss of a Phase-Separated Ternary Free-Radical/Cationic Hybrid System for the Application of Self-Matting Coatings

**DOI:** 10.3390/polym14122371

**Published:** 2022-06-11

**Authors:** Ingrid Calvez, Caroline R. Szczepanski, Véronic Landry

**Affiliations:** 1NSERC—Canlak Industrial Research Chair in Interior Wood-Products Finishes, Department of Wood and Forest Science, Université Laval, Québec, QC G1V 0A6, Canada; ingrid.calvez.1@ulaval.ca; 2Department of Chemical Engineering & Materials Science, College of Engineering, Michigan State University, East Lansing, MI 48824, USA; szcz@msu.edu

**Keywords:** surface wrinkle, self-matting, radical/cationic hybrid system, UV-curable coating, phase separation

## Abstract

Hybrid free-radical/cationic systems can generate phase-separated polymers or interpenetrating networks driven by photopolymerization. In this study, phase separation of a ternary mixture composed of a polybutadiene urethane diacrylate (PBUDA), a cycloaliphatic diepoxyde (CE), and hexanediol dimethacrylate (HDDMA) was investigated. Using systematic variations of the initial composition of the mixture, a miscibility phase diagram of the ternary mixture was established. Based on this diagram, a reactive copolymer (poly(butyl acrylate-co-glycidyl methacrylate) (PBGMA)) was introduced in a reference hybrid system to manipulate the crosslinking network, polymer morphology, and properties (e.g., roughness, gloss, strain at break, and glass transition temperature T_g_). When cured as a coating, the ternary hybrid system showed a depthwise gradient of epoxy conversion, and thereby developed a mostly cured skin above a viscous sublayer of uncured monomer. This skin can develop compressive stress due to the swelling from the diffusion of unreacted monomers beneath, and if the compressive stress is significantly high, wrinkles appear on the coating’s surface. This work highlights how both skin thickness and elastic modulus impact wrinkle frequency and amplitude. It was demonstrated that these wrinkle parameters can be manipulated in the ternary system by the addition of PBGMA. We also demonstrated that by employing UV irradiation and varying the PBGMA content, it is possible to engineer coatings that range from smooth surfaces with high gloss to wrinkled topographies with a very low associated gloss.

## 1. Introduction

Photopolymerization has been widely used for coatings, inks, and adhesives because of its inherent spatial and temporal control, versatility, energy efficiency, and environmental friendliness (i.e., low content of volatile organic compounds) [1,2,3]. The development of hybrid free-radical/cationic UV-curable systems is attractive, as distinct advantages associated with each constituent can be engineered for the final material. The monomer or oligomer mixtures used in free-radical/cationic hybrid photopolymerization typically contain two types of monomer species: (meth)acrylates and epoxides. (Meth)acrylates, which undergo free-radical polymerization, are very attractive due to their fast reaction rates and their chemical availability and versatility [4,5,6]. The epoxides, which undergo cationic polymerization, are attractive because they do not suffer from oxygen inhibition (a common challenge with free-radical chemistry), and also exhibit low toxicity and shrinkage [7,8].

Typically, in a hybrid polymerization system, free-radical and cationic polymerizations occur simultaneously to form an interpenetrating polymer network (IPN) [9,10,11]. IPNs often exhibit varying degrees of phase separation, which depends principally on the polymerization kinetics of the components and the compatibility of the employed polymers. The degree of phase separation and polymer incompatibility impact the overall performance of hybrid system, and large-scale phase separation is typically associated with a loss of performance (mechanical properties, interfacial adhesion, etc.) [12]. To improve and stabilize the interface between immiscible domains in a polymer mixture, a third component, with a chemical affinity for each initial polymer, can be added. This component can be a homopolymer [13,14,15] or a copolymer [16,17]. Due to the dual affinity for each domain, this species can migrate to the interface of a phase-separated polymer mixture and reinforce the biphasic interface by acting as a physical thread that binds the two phases together, and may even form a miscible blend [18,19], thus mitigating the reduced performance associated with gross incompatibility between domains.

UV-cured IPNs of hybrid systems have been widely reported in the literature, and the kinetics of network formation, mechanical properties, and surface morphology are usually the parameters that are investigated [10,11,20,21,22]. In some applications, the control of surface morphology is important. This is particularly relevant for strategies to reduce the gloss of formed materials [3], as gloss reduction can be achieved through the generation of surface patterns. For example, UV-curable coatings can exhibit low gloss surfaces when random wrinkling at their surface arises after UV irradiation. Due to this effect, a large amount of work has been carried out on the fabrication of different surface patterns based on surface instabilities [3,23,24,25,26,27,28,29,30]. From these studies, it has been established that wrinkle formation on polymer surfaces can be obtained using different strategies. Here, we focus on utilizing a depthwise gradient in curing to create and control wrinkled surfaces. Basu et al. pioneered the comprehensive theory of wrinkle formation in gradient-crosslinked films [31]. Their work describes how an initially liquid coating develops a gradient in crosslinking during curing and, thus, the mechanical properties have an associated gradient [31]. This phenomenon is mainly explained by the decrease in the UV intensity throughout the depth of a coating [25,32]. The top layer is crosslinked to a larger extent than the bulk of the film and, thus, behaves like a solid skin. Upon further crosslinking of the foundation, the stress produced in the bulk develops and stabilizes the wrinkle formation [33,34].

In the present work, the miscibility of a ternary mixture comprised of a polybutadiene urethane diacrylate (PBUDA), a cycloaliphatic diepoxyde (CE), and hexanediol dimethacrylate (HDDMA) was studied. In fact, we previously demonstrated the immiscibility of CE and PBDUA prior to photopolymerization, which was dependent on the ratio of these two species [35]. Here, HDDMA, which had complete miscibility with both CE and PBUDA, was added to ensure compatibility. Therefore, a hybrid ternary mixture was designed, and a reactive copolymer (poly(butyl acrylate-co-glycidyl methacrylate) (PBGMA)) was introduced to manipulate the crosslink density of the network in order to develop patterned microstructures. When cured, wrinkles formed in these systems due to the highly crosslinked epoxy networks, which produced a depthwise mechanical gradient during curing, and thereby developed a solid skin at the top of the film. Furthermore, the low polymerization rate associated with the HDDMA monomer contributed to the formation of a more viscous foundation of the film. The impact of PBGMA content was studied to control the final polymer properties and modulate the wrinkle morphologies. To investigate changes in thermomechanical properties (e.g., tan(δ) behavior, Young’s modulus, and strain at break) at room temperature, dynamic mechanical analysis was used. Depthwise gradients in the degree of conversion were investigated by confocal Raman microspectroscopy (CRM). Surface wrinkle morphologies were analyzed by 3D optical profilometry, and gloss was measured. Overall, we demonstrated self-wrinkling in UV-cured polymer films formed from a hybrid free-radical/cationic system. These wrinkles have controlled frequency and amplitude and can be leveraged for the development of self-matting UV-curable polymers.

## 2. Materials and Methods

The oligomer, monomers, and copolymer employed in this study are presented in Table 1.

The radical photoinitiator 2-hydroxy-2-methyl-1-phenyl-propan-1-one (HMPP, UV absorption peaks at 244, 278, and 322 nm, Irgacure 1173) and the cationic photoinitiator 4,4-dimethyl-diphenyl iodonium hexafluorophosphate (IODS, UV absorption peak at 267 nm, Omnicat 440—IGM resins) were obtained from Canlak (Daveluyville, QC, Canada) and MarkChem Inc. (Mirabel, QC, Canada), respectively. Details regarding the synthesis and characterization of the copolymer poly(butyl acrylate-co-glycidyl methacrylate) (PBGMA) are provided in the Appendix A.

### 2.1. Determination of the Phase Diagram for the Ternary Mixture and Solubility Parameters

In our previous study [35], we demonstrated the immiscibility of CE and PBDUA prior to photopolymerization, which was dependent on the ratio of these two species. Based on these findings, a third monomer was included (HDDMA) in the mixtures studied here, to ensure compatibility. HDDMA is a hydrophobic dimethacrylate monomer, which is completely miscible with both CE and PBUDA. A ternary phase diagram was determined by directly observing the miscibility as a function of composition of these three constituents prior to photopolymerization. The mixtures were considered miscible when, after stirring at 25 °C, the solutions appeared homogeneous and optically transparent. If the solutions showed a hazy aspect, the mixtures were considered immiscible. A few additional experimental solutions were tested to determine more precisely the immiscible region on the phase diagram. Additional details on the design of the experiments are available in Appendix A.

Solubility parameters, estimated using Hoftyzer and van Krevelen method based on the Flory–Huggins theory, were used to better understand the system’s thermodynamics and mixture instability. The solubility parameter of each monomer is related to the dispersion *δ_d_*, polar *δ_p_*, and hydrogen bonding forces *δ_h_*, and can be predicted using Equations (1)–(3) [36]. The total solubility parameter is determined from Equation (4). The differences in the *δ_t_*s values of the components of the mixture were used to determine their solubility/miscibility. If the difference in *δ_t_* between two compounds reached roughly 5 J^1/2^·cm^−3/2^, the mixture was defined as immiscible.
(1)$$\delta_{d}=\frac{\sum^{}F_{di}}{V}$$
(2)$$\delta_{p}=\frac{\sqrt{\sum^{}F_{pi}^{2}}}{V}$$
(3)$$\delta_{h}=\sqrt{\frac{\sum^{}E_{hi}}{V}}$$
(4)$$\delta_{t}=\sqrt{\delta_{d}^{2}+\delta_{p}^{2}+\delta_{h}^{2}}$$

In the above equations, the molar attractions *Fdi*, *Fpi*, and *Ehi* of each structural group were found in the literature [4,36], and the molar volume *V* was calculated by dividing the molecular weight by the density of each monomer.

### 2.2. Formulation and UV-Curing Processes

According to the miscibility investigation and gloss results, a system composed of 40 wt.% CE, 15 wt.% PBUDA and 45 wt.% HDDMA was examined as a reference. The concentration of the radical and cationic photoinitiators (PIs) was 1 wt.% and 3 wt.%, respectively. This PI ratio was chosen based on our previous study, which showed that at a high concentration of cationic PI, the epoxide part formed very rough surfaces, while the radical PI had little impact on the final degree of conversion [35]. The reference system was modified by incorporating 1, 2, 3, 5, 7, and 10 wt.% PBGMA, while maintaining the CE/PBUDA/HDDMA ratio.

Films of 50 μm were prepared on metal panels to investigate the extent of polymerization and the properties of the films. All mixtures were vigorously stirred by magnetic agitation prior to each application, and all products were applied using a calibrated 4-sided applicator (BYK Gardner, Columbia, SC, USA) at a speed of 3 m/min. For UV curing, a medium-pressure mercury lamp (Ayotte Techno-Gaz Inc., Lanaudière, QC, Canada) mounted over a conveyor belt was employed, and the lamp had an output in the range of 250 to 400 nm. Polymerization of all films was carried out under air and at room temperature. A two-stage curing regimen was employed to optimize the formation of surface wrinkles during photopolymerization. A first pass was performed at an intensity of 800 mW/cm². After 15 min of rest, a second pass was performed at an intensity of 1200 mW/cm². The conveyor speed was set to 5 m/min. Irradiation intensity was measured using a UV radiometer (Power Puck II, Electronic Instrumentation and Technology, Inc., Leesburg, VA, USA).

### 2.3. Dynamic Mechanical Analysis (DMA)

Dynamic mechanical analysis (DMA Q800, TA instruments, USA) was used to characterize the glass transition (T_g_) of UV-cured films in tensile mode at a frequency of 1 Hz. The storage modulus (*E*′) and loss modulus (*E*″)—representing the elastic and viscous behavior of the materials, respectively—as well as the loss factor (tan δ), defined as the ratio between *E*″ and *E*′, were determined as a function of temperature, using a constant heating rate of 3 °C min^−1^ from −20 °C to 200 °C. The samples had dimensions of 2.5 × 0.5 cm (length × width) and a thickness of 85–100 μm. Each film was run through two successive temperature ramps to avoid any post-curing phenomena. The storage modulus was used to calculate the crosslinking density (*ν_XL_*) of the formulations using Equation (5) [37]:(5)$$\nu_{XL}=\frac{E_{min}^{\prime }}{3*R*T}$$
where *ν_XL_* is the crosslinking density (mol.m^−3^), $E_{min}^{’}$ is the minimum storage modulus (Pa), *R* is the gas constant (8.314 J·mol^−1^·K^−1^), and T is the temperature at the minimum storage modulus (K).

Stress–strain curves were obtained using the DMA Q800 in tensile mode at 30 °C, with a force rate of 1 N/min from 0 to 18 N. Young’s modulus was estimated using the initial slope of the stress–strain curves.

### 2.4. Confocal Raman Microspectroscopy (CRM)

During photopolymerization, consumption of acrylate double-bonds leads to a decrease in the acrylate absorption band intensities measured by CRM. Similarly, during cationic photopolymerization, there is a decrease in the epoxy absorption band intensity. Here, CRM was used for the depth profiling of each film. This method enables the determination of the conversion (i.e., acrylate and epoxy functionalities) as a function of the nominal depth and could explain the skin formation phenomenon. Conversion (%) was calculated using Equation (6), where A_0_ is the absorbance before irradiation and A_t_ is the absorbance during irradiation at time *t*.

The absorption band associated with the acrylate groups was observed at 1636 cm^−1^ [38]. Its area was followed as a function of nominal depth within the limits of the sample thickness, and normalized against an internal standard in order to avoid data misinterpretation due to the loss of intensity with increasing depth, fluctuations in the laser intensity, or other external perturbations [39]. The C=O stretching peak at 1720 cm^−1^ was chosen for this purpose [40]. Similarly, epoxide ring conversion was monitored via the absorption band at 790 cm^−1^ [41]. An average of three spectra was recorded for each ratio in the uncured state to obtain values for *A*_0_.
(6)$$\%conversion=\frac{\left(A0-At\right)}{A0}\times 100$$

Analyses were performed using a SENTERRA II Raman microscope (Bruker Optics Inc., Billerica, MA, USA) equipped with a motorized table (Märzhäuser Wetzlar, Wetzlar, Germany). The excitation wavelength was obtained using an argon-ionized laser providing 100 mW light intensity at 785 nm. This wavelength was beyond the absorption range of the photoinitiator; therefore, there were no post-curing results from the CRM analysis. Spectra were recorded from 6 co-additions of 5 s of analysis each, giving 30 s of data acquisition for each step to obtain adequate experimental data. For all experiments, a 50× lens with a 0.75 numerical aperture (NA) (Olympus, Japan) was used. Theoretically, this optical configuration provides a depth resolution of 1.2 μm (≈1.22 λ/NA) [42,43]. A first set of spectra was recorded as a function of depth through the film, with increments of 10 µm between each acquisition, to measure the gradient conversion after the first and the second UV irradiation. Six independent measurements were taken at different locations on each sample. A second set of spectra were recorded using a 100× lens with a 0.25 μm numerical aperture in oil as a function of depth through the film, with increments of 2 µm between each acquisition, to measure the depth conversion gradient with more accuracy after the first UV irradiation.

### 2.5. Surface Characterization

Roughness measurements were taken using a Contour GT-I 3D optical profiler from Bruker (USA). For all samples, profiles were scanned over a length of 4 × 4 mm with a 20× objective. Three measurements were performed at different locations on each sample, and three samples per formulation were prepared. The roughness surface parameter *S_a_* was used to evaluate the surface roughness. No filters were used for data analysis; only a tilt correction was performed. The average parameter *S_a_* refers to the arithmetic mean of the absolute value of the height within a sampling area, as shown in Equation (7): (7)$$S_{a}=\underset{a}{\iint }\left|z_{\left(x,y\right)}\right|dxdy$$
where *z* is the height of the measured point in the coordinates *x* and *y* [44].

For samples with a wrinkled surface, a treatment was performed on the data. A Gaussian regression filter was applied to separate the waviness from the roughness. To consider the roughness close to zero, a long wavelength cutoff (L-filter) of 0.08 was applied.

The critical wavelength (*λ_cr_*) of the wrinkle pattern, as shown in Figure 1, can be predicted using a stress balance analysis for an elastic layer attached to a viscous sublayer, and can be calculated according to Equation (8) [31].

The minimum stress required for buckling is defined as the critical compressive stress, and the wavelength of the corresponding wrinkle pattern is defined as the critical wavelength.
(8)$$\lambda_{cr}=2\pi \;{\left[\frac{1}{12\left(1-\nu_{f}{}^{2}\right)}\right]}^{1/2}\frac{E_{f}{}^{1/2}}{\sigma_{0}{}^{1/2}}\;h_{f}$$

Here, *h_f_* is the thickness of the wrinkle skin, *E_f_* and *ν_f_* are the modulus and Poisson’s ratio of the elastic skin, respectively, and *σ*_0_ is the in-plane compressive stress on the skin. In this study *ν_f_* = 0.4 is assumed, following previous estimates for typical glassy-state thermosetting polymers [31,45]. Following Equation (8), *λ_cr_* has a stronger dependence on the thickness of the film than its modulus, and *λ_cr_* is not very sensitive to the Poisson’s ratio of the skin.

The specular gloss of the hybrid system was characterized using a micro-TRI-gloss meter from BYK (USA). According to ASTM D523-14 [46], the gloss was measured at 60° and 85°. Three measurements were performed on each sample, and three samples per formulation were prepared.

The transparency of the coating films was analyzed using a Varian Cary 50 UV–Vis spectrophotometer (USA). Films were deposited onto the sample pool and analyzed with a scanning wavelength ranging from 400 to 800 nm, at a medium scanning speed (600 nm/min). The films were analyzed after two passes under UV light and 4 days of dark storage.

Burnish resistance was evaluated using a modified version of the standardized test method ASTM 6736 [47]. Coatings with a thickness of 50 μm applied on aluminum Q-panels were placed on the Elcometer 1720 Abrasion and Washability Tester (Elcometer Inc., Warren, MI, USA). Cheesecloth (4-ply, medium weave) was then deposited on the coating surface. To ensure that a reproducible and sufficient force was applied to the coating, a 500 g weight was placed over the tested samples. Abrasive pads were rubbed back and forth on the coating’s surface for 20 cycles. Gloss units at 85° were recorded with a BYK micro-TRI-gloss meter (BYK Gardner, USA) before and after burnishing. The gloss modification was measured by the difference in the gloss after and before burnishing.

Direct and reverse impacts were tested according to ASTM D2794 [48]. The impact tester (Elcometer 1615, USA) was used with a 2 lb (0.91 kg) weight with the hemispherical punch. The test was carried out on the coated side (direct impact) and the uncoated side (reverse impact) of the steel plates. The coated steel plate was placed under the falling guide, and the height was gradually increased until the coating broke under the impact. The value of the maximum height where the coating resisted the impact was used to characterize the direct or reverse impact resistance.

## 3. Results and Discussion

### 3.1. Miscibility Studies of a Ternary Free-Radical/Cationic Mixture

Co-incident phase separation and polymerization are dependent on both the thermodynamics and the reaction kinetics of the given hybrid system. Specifically, thermodynamics dictates the degree of incompatibility between monomers and monomer–polymer solutions and, thus, the driving force for phase segregation. According to our previous study, CE and PBUDA show incompatibility in their monomer liquid states, depending on their ratio, which leads to the formation of a heterogeneous and totally phase-separated network [35].

Based on these prior results, a third monomer was introduced to increase the miscibility between CE and PBUDA. Specifically, HDDMA—a hydrophobic monomer—was selected due to its total miscibility with both CE and PBUDA. Table 2 shows the solubility parameters for each monomer, obtained using Equations (1)–(4) [36].

The immiscibility between two compounds starts when the difference in their total solubility parameters (*δ_t_*) reaches roughly 5.0 J^1/2^·cm^−3/2^ [4,36]. The immiscibility between PBUDA and CE is thus explained, as the difference in solubility parameters between these two species is 6.0 J^1/2^·cm^−3/2^. On the other hand, PBUDA and HDDMA show good miscibility, which is expected based on the difference of 1.3 J^1/2^·cm^−3/2^ in their solubility parameters. Lastly, a difference of 4.7 J^1/2^·cm^−3/2^ between HDDMA and CE indicates miscibility. However, given that this value is very near to the 5.0 J^1/2^·cm^−3/2^ threshold, it could possibly lead to phase separation under suitable photopolymerization conditions and monomer ratios. A study of the miscibility of the ternary mixture prior to UV curing was carried out at 25 °C using a design-of-experiments (DoE) plan, which is available in the Appendix A (Appendix A). The results of these miscibility experiments and the corresponding ternary phase diagram are presented in Figure 2.

As shown in Figure 2, PBUDA and CE can be soluble in HDDMA, depending on their ratio. In particular, miscible mixtures with high HDDMA monomer concentrations have an associated low viscosity. The location corresponding to the composition prior to photopolymerization of the mixture explored in detail throughout this study is indicated by a red star. This mixture was chosen because of its good miscibility in the monomers’ liquid state and the gloss values of the cured films (discussed later). A block copolymer composed of n-butyl acrylate and glycidyl methacrylate chains (PBGMA) was introduced at different concentrations to manipulate the network architecture of the formed material, e.g., crosslinking density and miscibility (T_g_). The formulations investigated and their associated viscosities are presented in Table 3.

With the experimental formulations (Table 3), DMA analysis was first performed to characterize the glass transition temperature and the crosslinking density after photopolymerization of these mixtures (Figure 3).

As predicted by the solubility parameters, the initially miscible mixtures became a phase-separated network after photopolymerization, as indicated by multiple local maxima in the tan δ profiles, indicating the formation of phase-separated domains [49,50]. For reference, the associated T_g_ values of the neat epoxy (CE) and 1:1 HDDMA:PBUDA were approximately 178 and 86 °C, respectively. Two local maxima in the tan δ profile were observed for the reference mixture (0 wt.% PBGMA) at 86 and 132 °C, corresponding to an acrylate-rich phase and an epoxy-rich phase, respectively. When 5 wt.% PBGMA was introduced into the resin mixture, the tan δ profile shifted to lower temperatures, with two local maxima at 69 and 122 °C. This significant shift in tan δ can be explained by the presence of the flexible butyl acrylate part of the copolymer, which has an associated low T_g_ (−54 °C). With 10 wt.% loading of PBGMA, a single tan δ peak with one clear maximum was observed at 92 °C, indicating a decrease in the heterogeneity of the polymerized material. At this loading level of PBGMA, the copolymer fraction was large enough that it acted as a reactive compatibilizer between the (meth)acrylate and epoxy phases, with the butyl acrylate having an affinity for the (meth)acrylate part and the glycidyl epoxy group being able to react with the epoxy groups of the resin [51]. All the T_g_ values and the corresponding crosslinking density for each sample are presented in Table 4.

As shown in Table 4, with the addition of PBGMA, the crosslinking density decreased, which was correlated with the decreased T_g_s. This is not surprising, as the copolymer acts as a spacer between the polymer chain and extends the length between crosslinking points where the epoxy rings in PBGMA might react with the epoxy part (CE) [52]. With the significant changes observed in tan δ behavior, T_g_s, and crosslinking density due to the incorporation of PBGMA, it was expected that the mechanical properties and surface morphologies of the formed films would also vary significantly with PBGMA content.

Micro-phase-separated morphology, which was expected based on the multiple maxima observed in the tan δ profile, was supported by the SEM cross-section images of the 5 wt.% PBGMA film, as shown in Figure 4. For this analysis, the film was fractured in liquid nitrogen, and the fractured surface was examined.

### 3.2. Wrinkling of Structures by Skin Formation on a Viscous Sublayer

Depending on the PBGMA content, different surface morphologies were observed, from smooth to wrinkled structures. Basu et al. proposed a hypothesis on the mechanism of wrinkle formation. They explained that during polymerization, a liquid coating develops a depthwise gradient in the degree of crosslinking (e.g., solidification), which corresponds to a gradient in physical properties, such as diffusivity, modulus, etc. [31]. The top layer, being the most cured, develops a significant modulus prior to the remainder of the film, and behaves like a solid skin, with the ability to support stresses. The depthwise gradient in the degree of crosslinking also results in a gradient in unreacted oligomer concentration in the opposite direction, which causes the oligomer to diffuse upward into the skin. 

Figure 5 shows the effect of PBGMA content on the surface morphology. Surface topography was characterized after two passes under the UV light, with a 15 min rest period between each pass. During this rest time, the surface wrinkles developed. The influence of this curing process is explained in the next section. As seen in Figure 5a, the reference system without any PBGMA did not display any wrinkling, and the surface was relatively smooth. However, with 3 wt.% or more PBGMA included in the resin formulation, wrinkles appeared (Figure 5b–e). The wrinkle patterns generated lacked any directional order, and the amplitude and frequency of the wrinkles varied according to the percentage of PBGMA. As shown in Figure 5, the amplitude and frequency were the highest at 5 wt.% (c), and then decreased at 7 wt.% (d) and 10 wt.% (e) PBGMA.

Raman analyses were performed as a function of the film thickness to better understand the formation of surface wrinkles. The conversion profiles for the acrylate and epoxy parts as a function of depth and PBGMA content are shown in Figure 6.

On the one hand, the epoxy part showed a high depthwise gradient in conversion within the film thickness, which was apparent even after the second pass of UV irradiation. After the first pass under UV light (Figure 6A), the epoxy conversion reached 45 to 53% at the top of the film, compared to 17 to 27% at the bottom, depending on PBGMA content. On the other hand, the (meth)acrylate conversion (Figure 6B) was rather consistent throughout the film thickness, with a relatively low average conversion close to 20%. This low conversion is characteristic of a methacrylate system, with a rate of polymerization rate much lower than its acrylate equivalent, due to the formation of stable ternary radicals [53].

Two phenomena can explain the depthwise gradients in the extent of conversion. First, Song et al. explained that UV irradiance is attenuated through coatings, due to the absorption by the photoinitiator (PI) [25]. The Beer–Lambert law describes the effect of thickness and coating composition on light absorbance (Equation (9)):(9)$$I_{a}=I_{0}\left(1-10^{-\varepsilon cl}\right)$$
where *I_a_* and *I*_0_ are the absorbed and incident irradiance of light, respectively, *ε* is the molar absorption coefficient of the absorber (i.e., PI), *c* is the absorber’s concentration, and l is the film thickness. The authors demonstrated that with PI absorption, the variation in the irradiance throughout the thickness leads to a conversion gradient. The surface of the coating receives a higher light irradiance and, hence, has a higher conversion. In some cases, this conversion at the surface leads to the formation of a solidified “skin” layer. This skin can develop compressive stress due to swelling of unreacted monomers that diffuse from beneath the skin layer. When the compressive stress is high enough to make the skin buckle, wrinkles appear on the coating’s surface. UV-curable epoxy resins have a tendency to develop a depthwise conversion gradient, as the top cures more rapidly [25].

The other factor contributing to the depthwise gradients in these hybrid films is their phase-separated morphologies, which result in an overall hazy appearance, with lower transmittance due to the mismatch in refractive index between phase domains. This could also impact the penetration of UV rays into the film and accentuate the depthwise gradient phenomenon. Figure 7 shows the impact of PBGMA content on transmittance in fully cured hybrid systems. The lowest transmittance was observed with systems containing 3, 5, and 7 wt.% PBGMA, compared to the reference system and the one containing 1 wt.% PBGMA. On the one hand, light transmittance is a result of the formation of co-continuous soft/hard phase-separated and light-scattering domains. Interestingly, the mixture with 10 wt.% PBGMA showed a higher transmittance than those containing 3, 5, or 7 wt.% PBGMA. This increase in light transmittance could be associated with a reduction in the size and sharpness of the domain interfaces, along with the formation of less distinct phases, both of which would limit the degree of visible light scattering [50]. On the other hand, as shown in Figure 5, no wrinkles were observed below 3 wt.% PBGMA. However, with the addition of 3 wt.% PBGMA or more, wrinkles appeared. The decrease in transmittance was attributed to the formation of micrometer-sized wrinkle patterns, which caused scattering of incident light [54]. The lowest transmittance corresponded to the system with the highest wrinkle amplitude, which was the mixture with 5 wt.% PBGMA.

The combination of depthwise gradient in the curing of the epoxy part and the low conversion of the methacrylate part strongly contributes to the formation of surface wrinkles in UV-cured films. In fact, the unreacted low-molecular-weight monomers below the hard skin diffuse upwards into the oligomer-depleted crosslinking skin.

The second stage of UV curing at a higher light intensity provides additional energy to complete the polymerization of both the (meth)acrylate and epoxy parts, and fixes the patterns. As shown in Figure 6, after a second pass under UV light at higher intensity, the epoxy conversion increased to 80%, but the depthwise gradient was still observed. Regarding the (meth)acrylate part, the conversion increased significantly to 70% after the second pass. Additionally, cationic polymerization is known to continue to polymerize due to its dark-living characteristics. This post-curing behavior was investigated via FTIR after 4 days of dark storage at room temperature. The measures taken at the top and bottom of the cured film are available in the Appendix A. Conversion increased during this storage period, and reached a final level of 96–98%.

### 3.3. Gloss and Surface Wrinkling/Roughness

Based on the observed surface morphologies after curing (Figure 5), surface roughness (S_a_)—which considers both waviness and roughness—was analyzed, and the gloss was measured for each system at various wt.% PBGMA. The results are summarized in Figure 8.

As shown in Figure 8, the surface roughness increases significantly up to 5 wt.% PBGMA, reaching a value of 3.8 μm. With increased loading of PBGMA, the roughness decreases. At the same time, the gloss also decreases significantly with PBGMA loading up to 5 wt.%, reaching a value of 4.7 GU at 85°, and then increases again with additional PBGMA. These data are consistent with the rough structure observed and presented in Figure 5. Unexpectedly, it was noted that between 0 and 2 wt.% PBGMA, the roughness increased even without observing wrinkles on the film surface. Here, the roughness was provided by microvoid formation (see Appendix A) due to phase separation during the photopolymerization. The difference in curing shrinkage [55,56,57] between (meth)acrylate (around 20%) and epoxy resins (around 5–10%) during photopolymerization leads to internal volume rearrangements to compensate for the difference in shrinkage, and could explain this roughening. Additionally, using a large amount of HDDMA as a reactive monomer may contribute to this microvoid formation due to the photocuring shrinkage phenomenon [58].

To measure wrinkle frequencies, images of the wrinkle patterns were obtained by optical profilometry and analyzed using R software. Smoothing was performed on the raw data to reduce the roughness and, thus, measure only the waviness. An average of 100 random X profiles was performed; the results are presented in Figure 9B. The wrinkle amplitudes were obtained by applying a Gaussian filter with Bruker-Vision 64 software (Figure 9A).

No wrinkles were observed at 0, 1, and 2 wt.% PBGMA. However, with the addition of 3 wt.% PBGMA or more, wrinkles appeared. The amplitude increased up to 5 wt.% loading, and then decreased at higher PBGMA contents. This tendency was also verified by surface roughness measurements (Figure 8). According to Equation (8), it is possible to predict the critical wavelength (*λ_cr_*) of the wrinkle patterns as a function of skin modulus (*E_f_*), thickness (*h_f_*), Poisson’s ratio (*ν_f_*), and compressive stress (*σ*_0_) [29]. The minimum stress required for wrinkling is defined as the critical compressive stress, and the wavelength of the corresponding wrinkle pattern is defined as the critical wavelength [31]. In systems with a viscous sublayer, the viscosity or thickness of the viscous sublayer does not have any effect on *λ_cr_*; however, it does determine its growth rate [59,60,61]. In addition, the critical wavelength shows a stronger dependence on the skin thickness. 

To obtain a quantitative measure of the skin thickness (*h_f_*), depthwise CRM measurements were performed after the first UV irradiation pass. This technique allowed us to obtain the thickness of the cured film by measuring the conversion in function of the thickness, and incremental measurements of 2 μm were made. An observed abrupt decrease in the epoxy conversion in the film thickness was used to determine *h_f_*, as shown in Figure 10, and the results are presented in Table 5.

The frequency also depends on the Young’s modulus of the skin (*E_f_*). Here, an approximation was made—*E_f_* was calculated according to the stress–strain curves obtained by DMA (SI—Appendix A). The films were tested after two passes of UV irradiation and 4 days of dark storage. Young’s modulus thus corresponds to fully cured films. Furthermore, upon releasing the strain, the skin experienced a compressive stress [62]. This compressive stress, once above a critical value, was relieved by out-of-plane deformation, creating wrinkles [63]. The compressive stresses *σ*_0_ in wrinkling coatings can be easily estimated using Equation (8), where λ is the experimental measurement of the wrinkle wavelength. The results are shown in Table 5. The wrinkle amplitude increases with the magnitude of the stress [27,64]. In fact, the wrinkle amplitude is given by
$\frac{A}{\lambda_{cr}}\sim \varepsilon_{0}^{1/2}$
, where *λ_cr_* is the critical wavelength and *ε*_0_ is the applied global strain, which is related to the compressive stress *σ*_0_ by $\sigma_{0}=E\varepsilon_{0}/\left(1-\nu \right)$ [27,61]. The highest amplitude was observed for the 5 wt.% PBGMA system, for which the skin thickness and compressive stress were the highest. Furthermore, when modifying the crosslinking network with a higher amount of PBGMA, the elastic modulus and the wrinkles’ wavelength decreased, while the wrinkles’ amplitude increased.

According to Basu et al., coatings with highly elastic moduli generate large-wavelength, low-amplitude wrinkles that develop slowly; in other words, these coatings are the least likely to wrinkle [60]. Coatings containing 0 to 2 wt.% PBGMA showed the highest crosslinking density and elastic moduli, which directly impact the growth rate of wrinkles. It is possible that the 15 min rest between the two UV irradiation passes was not enough to develop wrinkles in these systems. In addition, the curing time (or UV energy dose) may also be a parameter that impacts the surface wrinkle formation. In fact, increasing the curing time increases the conversion of both epoxy and (meth)acrylate parts; thus, the swelling of unreacted acrylates that diffuse beneath the skin layer decreases. This, in turn, decreases the amplitude of the wrinkles formed. However, decreasing the curing time reduces the conversion of both parts, thus decreasing the thickness of the skin layer.

### 3.4. Mechanical Characterizations

Several mechanical analyses were performed to characterize the impact of PBGMA on the ternary hybrid coating. Stress–strain curves were collected by DMA, and are presented in the Appendix A.

Direct and reverse impact resistance are mechanical parameters for the evaluation of coating properties. The flexibility of the coatings was characterized using reverse impact testing, and the results are presented in Table 6. These results indicate that the amount of PBGMA does not influence the flexibility. The brittle behavior of the formed coatings is attributable to the large epoxide fraction in the hybrid systems. The impact resistance, evaluated by drop-weight impact testing, decreased with the addition of PBGMA. The decreased crosslink density prevents the polymer from distributing and absorbing stress evenly. Thus, the material can absorb less stress and energy before failure. These results are consistent with the stress–strain curves (Appendix A). 

In practical applications, wrinkled surfaces are often exposed to a variety of complex and harsh environments, and may lose their low gloss over time due to the damage caused by various mechanical external forces, such as burnishing. Therefore, burnish resistance was studied to determine the stability of the surface structures and, therefore, the gloss. As shown in Table 6, for systems with smooth surfaces, the increase in gloss after burnishing was greater. For example, the gloss increased by two units after burnishing for the reference system, while the gloss remained almost unchanged after burnishing for the system containing 5 wt.% PBGMA, which was composed of a wrinkled surface. Thus, the wrinkle structures have a good resistance to burnishing, and maintain low gloss. In fact, it is possible that the increase in surface roughness decreases the coefficient of friction due to the reduction in the contact area, which significantly reduces surface burnishing [65,66].

## 4. Conclusions

This study investigated the miscibility of an initially incompatible hybrid free-radical/cationic system by adding a methacrylate monomer. A ternary phase diagram aided in the selection of a reference mixture composed of CE/PBUDA/HDDMA, which was miscible before irradiation. A reactive copolymer composed of n-butyl acrylate and glycidyl methacrylate (PBGMA) was introduced into the ternary hybrid system. We investigated the impact of PBGMA content on the crosslinking network manipulation, as well as the polymer morphology and properties. CE/PBUDA/HDDMA showed phase separation during irradiation, as indicated by two local maxima in the tan δ profile, attributed to acrylate-rich and epoxy-rich phases. The addition of PBGMA decreased both T_g_s based on the presence of a flexible chain with low T_g_. At 10 wt.% PBGMA, the ternary mixture showed only one tan δ peak, which was attributed to the butyl acrylate moiety having an affinity for the (meth)acrylate part and the glycidyl epoxy group being able to react with the epoxy groups of the resin. In addition, crosslinking density (*ν_XL_*) enabled us to understand the impact of the PBGMA in the system. With the addition of PBGMA, both T_g_s and *ν_XL_* decreased, as the copolymer acted as a spacer between the crosslinking points. Furthermore, under UV irradiation, and depending on PBGMA content, the system showed wrinkled surfaces, which significantly decreased the gloss. Overall, the ternary hybrid system showed a depthwise gradient of epoxy conversion. Indeed, the top layer, mostly cured, emerged as a skin that could develop wrinkle patterns. The sublayer was composed of a viscous medium of uncured monomers, as (meth)acrylate exhibited low conversion (20%) after the first irradiation. This sublayer diffused up into the skin and produced an elastic compressive stress in the skin, creating wrinkles. Finally, this work demonstrates that both skin thickness and elastic modulus have an impact on wrinkle frequencies and amplitudes with the addition of PBGMA. Increasing the amount of copolymer initially increased the amplitude of the wrinkles and, thus, decreased the gloss; then, after reaching an optimal level at 5 wt.% PBGMA with the lowest gloss of 4.7 GU at 85°, the amplitude decreased, and the gloss increased again. Furthermore, the wrinkled surfaces showed a good resistance to burnishing, which did not alter the gloss. Finally, the manipulation and control of surface morphologies could help improve current formulations for the next generation of self-matting UV-curable coatings.

## Figures and Tables

**Figure 1 polymers-14-02371-f001:**
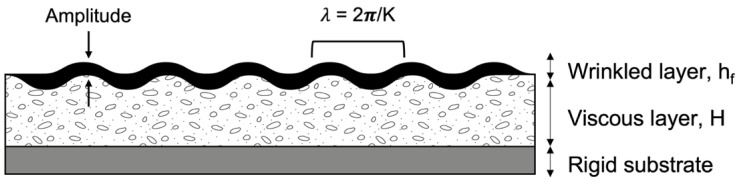
Schematic illustration of a compressed film on a viscous layer in a wrinkled state.

**Figure 2 polymers-14-02371-f002:**
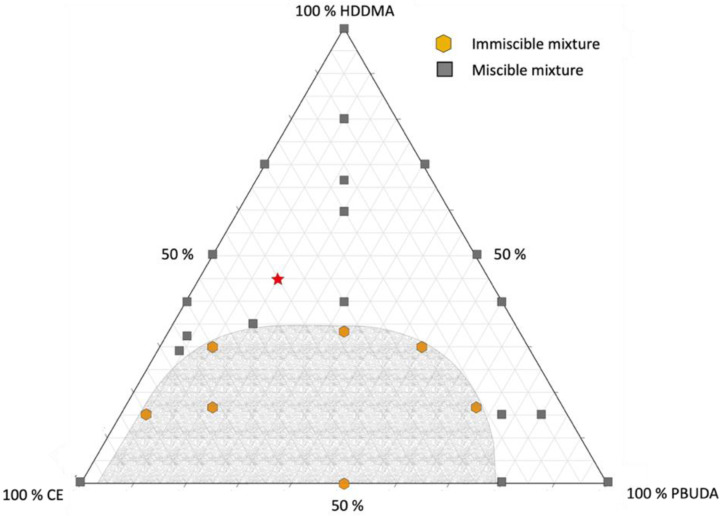
Ternary phase diagram of CE/PBUDA/HDDMA in a liquid state at 25 °C. The shaded region indicates immiscibility, and the star represents the experimental composition selected for further analysis in this study.

**Figure 3 polymers-14-02371-f003:**
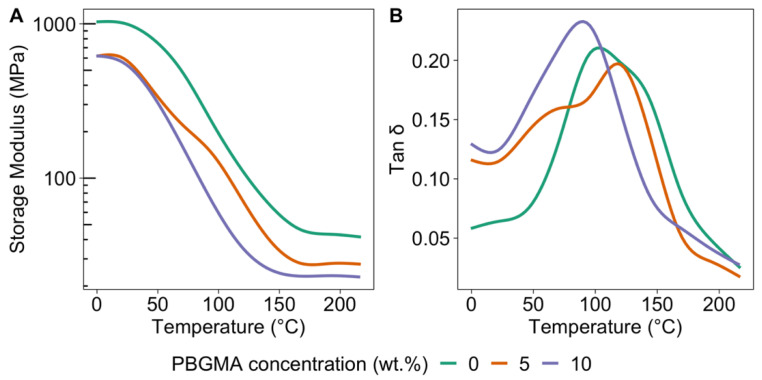
(**A**) Storage modulus and (**B**) tan δ curves of the reference mixture, 5 wt.%, and 10 wt.% PBGMA UV-cured films as a function of temperature, obtained by DMA.

**Figure 4 polymers-14-02371-f004:**
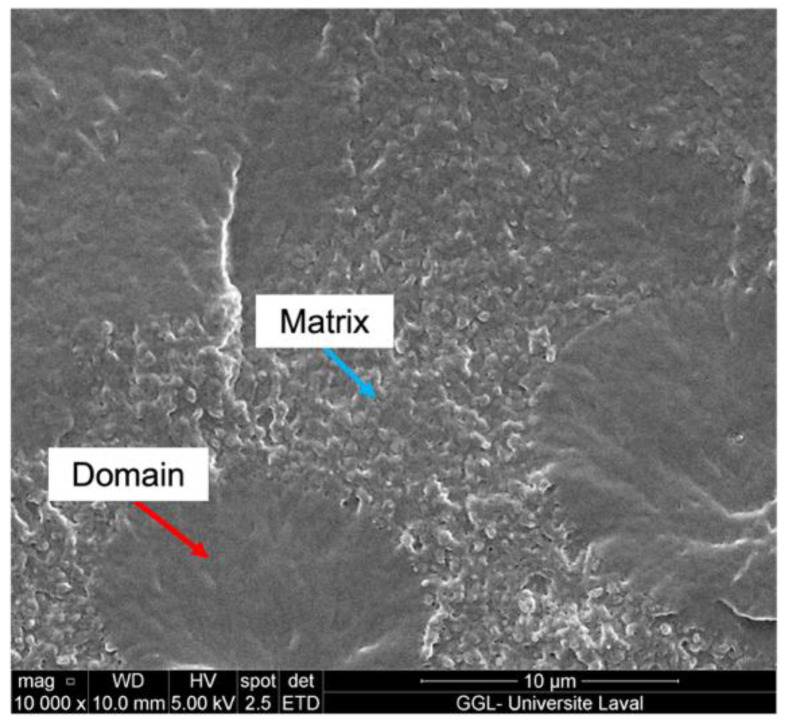
SEM cross-section images of the film prepared from the 5 wt.% PBGMA.

**Figure 5 polymers-14-02371-f005:**
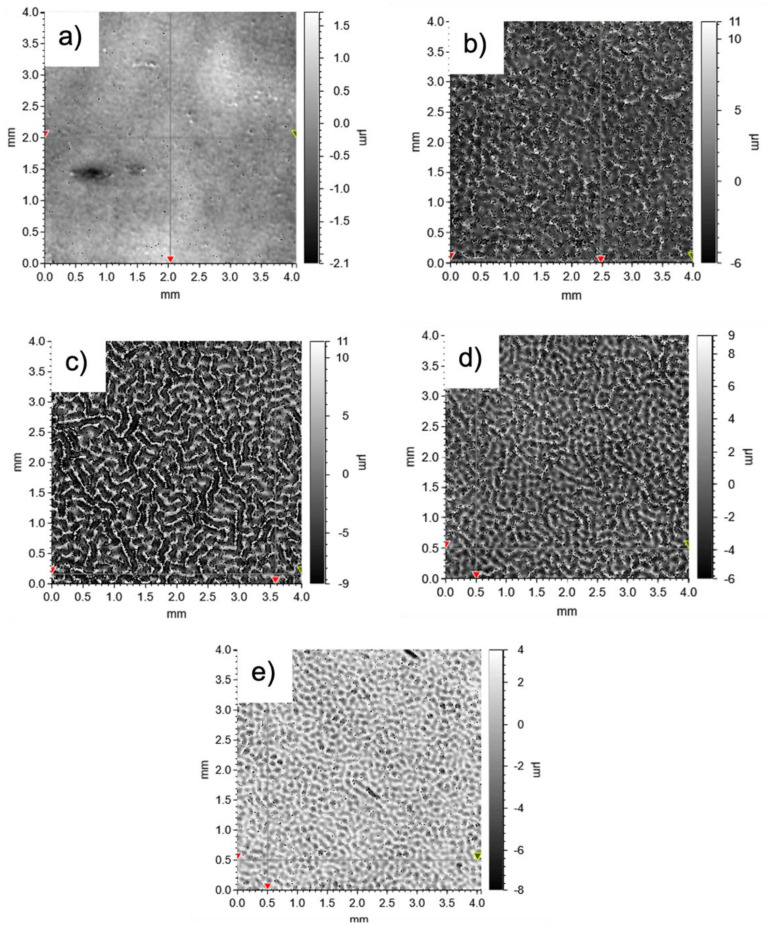
Images obtained by 3D optical profilometry of (**a**) the reference system, and (**b**) 3 wt.%, (**c**) 5 wt.%, (**d**) 7 wt.%, and (**e**) 10 wt.% PBGMA.

**Figure 6 polymers-14-02371-f006:**
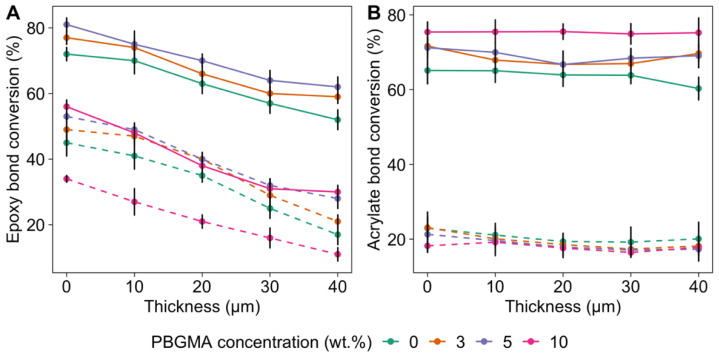
Conversion of (**A**) the epoxy part and (**B**) the acrylate part as a function of depth and PBGMA concentration. Raman spectra were recorded every 10 µm of depth. The dashed lines correspond to the conversion after a first pass under UV light at 800 mW·cm^−2^, and the solid lines to the conversion after a second pass under UV light at 1200 mW·cm^−2^. The values presented are averaged from six measurements.

**Figure 7 polymers-14-02371-f007:**
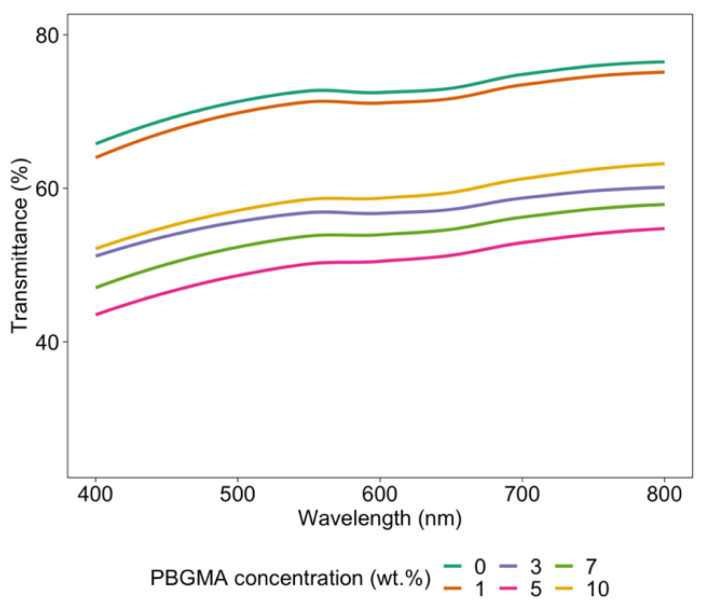
Transmittance curves obtained by UV–Vis spectrometry at various PBGMA contents after 2 passes under UV irradiation.

**Figure 8 polymers-14-02371-f008:**
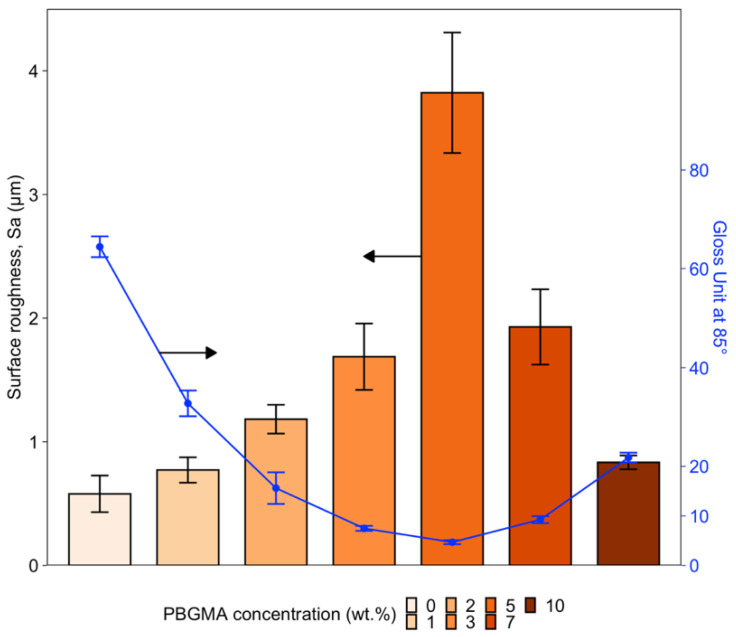
Impact of PBGMA concentration on surface average roughness (Sa) and gloss at 85°. Three samples per formulation were analyzed, and the values presented were averaged from three measures per sample.

**Figure 9 polymers-14-02371-f009:**
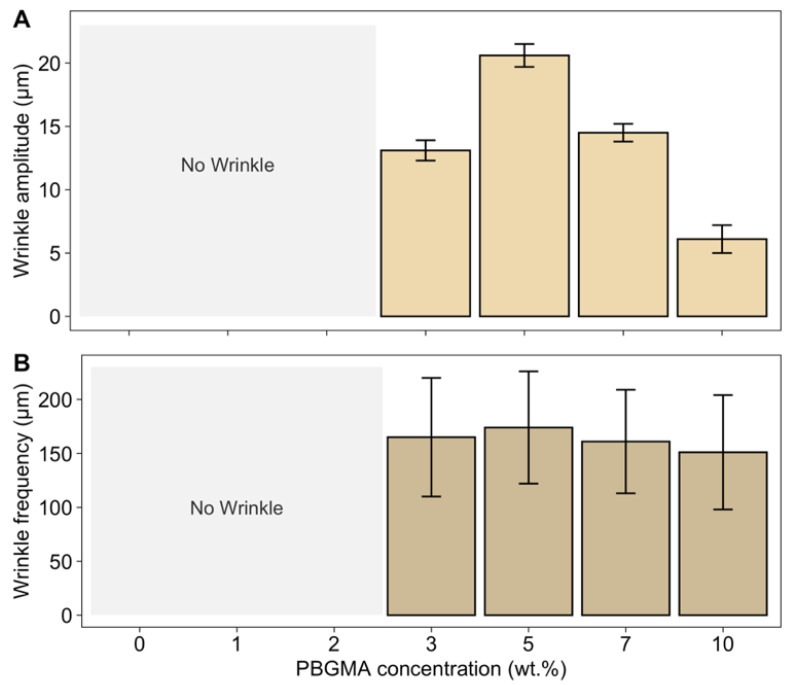
(**A**) Wrinkle amplitudes and (**B**) frequencies according to PBGMA concentration. Three samples per formulation were analyzed. The amplitude values are averaged from three measurements and the frequency values from an average of 100 random X profiles.

**Figure 10 polymers-14-02371-f010:**
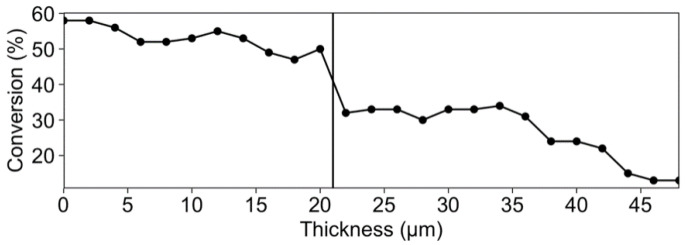
CRM analysis of epoxy conversion as a function depth for the 5 wt.% PBGMA system.

**Table 1 polymers-14-02371-t001:** Chemical and commercial names, chemical structures, and suppliers of the UV-curable chemicals used in the study.

Chemical and Commercial Name	Chemical Structure	Supplier
Polybutadiene urethane diacrylate (PBUDA)—Dymax BR-640D	N.A.	EMCO-Inortech (Terrebonne, Canada)
1,6-hexanediol dimethacrylate (HDDMA)—Miwon Miramer M201	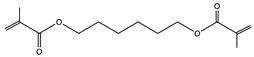	EMCO-Inortech (Terrebonne, Canada)
Epoxycyclohexylmethyl 3*,*4*-*epoxycyclohexanecarboxylate (CE)—Omnilane 1005	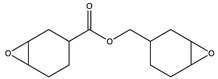	MarkChem Inc. (Mirabel, Canada)
Poly(butyl acrylate-co-glycidyl methacrylate) (PBGMA)	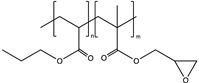	Information in the Appendix A

N.A.: Not Available

**Table 2 polymers-14-02371-t002:** Solubility parameters calculated for each monomer in this study.

	PBUDA ^1^	HDDMA	CE
*V* (cm^3^·mol^−1^)	507	255	215
*δ_d_* (J^1/2^·cm^−3/2^)	12.7	15.8	20.8
*δ_p_* (J^1/2^·cm^−3/2^)	5.0	3.8	3.5
*δ_h_* (J^1/2^·cm^−3/2^)	9.3	7.4	7.8
*δ_t_* (J^1/2^·cm^−3/2^)	16.5	17.8	22.5

^1^ Molecular weight obtained by MALDI-TOF

**Table 3 polymers-14-02371-t003:** Composition of the reference formulation and the formulations modified with various concentrations of PBGMA, and their associated viscosities measured at 25 °C.

	CE	PBUDA	HDDMA	PBGMA						
Composition (wt.%)	40	15	45	0	1	2	3	5	7	10
Viscosity (mpa·s) at 25 °C	180–450	6000 at 60 °C	1–10	51 ± 2	59 ± 1	76 ± 5	104 ± 6	139 ± 3	207 ± 4	234 ± 5

**Table 4 polymers-14-02371-t004:** T_g_(s) of the (meth)acrylate and epoxy domains, as well as overall crosslinking densities obtained via DMA. Data are presented for the reference ternary mixture (0 wt.% PBGMA) and samples with varying wt.% of PBGMA (all mixtures contained 1 wt.% radical photoinitiator and 3 wt.% cationic photoinitiator). The T_g_s of the (meth)acrylate (1:1 PBUDA:HDDMA) and the neat epoxy was 86 and 178 °C, respectively.

	PBGMA (wt.%)						
	0	1	2	3	5	7	10
Tg_1_ (°C) Acrylate part	95 ± 1	88 ± 2	82 ± 1	70 ± 4	65 ± 1	68 ± 3	91 ± 1
Tg_2_ (°C) Epoxy part	129 ± 1	126 ± 3	132 ± 1	130 ± 1	124 ± 2	120 ± 2	
*ν_XL_* (mol·m^−3^)	3690 ± 110	3420 ± 50	3260 ± 50	2830 ± 110	2560 ± 90	2310 ± 90	1980 ± 60

**Table 5 polymers-14-02371-t005:** Measured wrinkle wavelength, estimated Young’s modulus, skin thickness, and compressive stress of the systems with various PBGMA concentrations.

PBGMA (wt.%)	0	1	2	3	5	7	10
λ_measured_	/	/	/	165	174	161	151
*E* (MPa)	8.4	8.1	7.5	6.9	5.7	5.1	4.2
*h_f_* (μm)	6	10	10	12	20	14	6
*σ*_0_ (MPa)	/	/	/	0.14	0.29	0.15	0.03

**Table 6 polymers-14-02371-t006:** Direct and reverse impact, and burnish resistance of each film, at different PBGMA concentrations.

PBGMA Concentration (wt.%)	Reverse Impact (kg/cm)	Direct Impact (kg/cm)	Gloss before Burnishing (at 85°)	Gloss after Burnishing (at 85°)	Difference in Gloss
0	8.0 ± 0.0	43.0 ± 1.0	62.4	64.3	1.9 ± 0.9
1	8.0 ± 0.5	43.0 ± 1.0	25.7	27.3	1.6 ± 0.4
2	8.0 ± 0.5	33.0 ± 1.0	12.4	14.9	2.5 ± 0.3
3	8.0 ± 0.5	31.0 ± 1.0	7.0	8.7	1.7 ± 0.3
5	8.0 ± 0.5	33.0 ± 1.0	4.3	4.9	0.6 ± 0.2
7	8.0 ± 0.5	33.0 ± 1.0	8.6	9.0	0.4 ± 0.5
10	9.0 ± 0.0	32.0 ± 1.0	20.9	22.8	1.9 ± 0.1

## Data Availability

The data presented in this study are available on request from the corresponding author.

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
