# Peer review of "Effect of Copolymer on the Wrinkle Structure Formation and Gloss of a Phase-Separated Ternary Free-Radical/Cationic Hybrid System for the Application of Self-Matting Coatings"

_polymers, 2022, doi:10.3390/polym14122371_

Round 1

Reviewer 1 Report

Hybrid free radical/cationic UV-curable systems have distinct advantages which can engineer each constituent to final material. In this work, the authors designed a hybrid ternary mixture for UV-curing coating and introduced a reactive copolymer (poly (butyl acrylate-co-glycidyl methacrylate), PBGMA) to regulate the crosslinking density and surface wrinkle. By added hexanediol dimethacrylate (HDDMA) to ensure the miscibility of cycloaliphatic diepoxyde (CE) and polybutadiene urethane diacrylate (PBUDA), interpenetrating polymer network (IPN) of hybrid systems are constructed to develop the self-matting UV-curable polymer. Moreover, the contents of PBGMA can be controlled to regulate the morphology of wrinkles. This work can be recommended for publishing in Polymers after some minor revisions. Some comments are listed below:

(1) The formation of wrinkle in this work is explained as the depth-wise gradient in the degree of crosslinking. The wrinkle developed in the rest time after first exposure, and there is higher conversion for epoxy and lower conversion for acrylate according to Fig.6 in this manuscript. So, the formation of wrinkle is more like the result of different reaction speed about epoxy part and acrylate part. Less-reacted acrylate monomer diffuse to epoxy solid skin resulting the swell and wrinkle. The real reason of self-wrinkling behavior should be confirmed.

(2) Why this process of UV-curing need exposure twice? Would the morphology or mechanical properties change with only one time exposure at intensity of 1200mW/cm2?

(3) There is no chapter 3.2.

(4) Should the study about wrinkle formation (Fig.5) and conversion of the epoxy part (Fig.6) be put in “3.3. Gloss and Surface Wrinkle/Roughness”? The first part is discussing the miscibility of mixture and phase separated, and the epoxy bond conversion and wrinkle formation seems the result of depth-wise gradient in the degree of crosslinking.

Reviewer 2 Report

This paper describes the surface morphology of wrinkles structures produced by the phase-separated polymers.  While this study makes some contributions to the fields of polymer chemistry and physics concerning the microphase separation of copolymers, I have some reservations that the authors should address before publication.  My comments are as follows.

1.     The authors should show the chemical structures of the compounds in the text rather than in the supplementary.  They also need to add a schematic procedure for the wrinkles structures synthesis.

2.     The photo radical/cationic polymerization of glycidyl methacrylate has already been reported (polymers, 2012, 4, 1580).  The authors should add this reference and discuss it in the introduction.

3. Figure 5…Does the wrinkle appearance depend on the cure time?  The authors need to investigate variation in the surface morphology with curing time.

4.     Figure 7…The investigation of the PBGMA content on transmittance does not provide systematic results concerning their correlation.  The authors should explain why the transmittance arranges in this order of PBGMA content.

Round 2

Reviewer 2 Report

The revised manuscript is acceptable.